# Uranium Removal from Aqueous Solutions by Aerogel-Based Adsorbents—A Critical Review

**DOI:** 10.3390/nano13020363

**Published:** 2023-01-16

**Authors:** Efthalia Georgiou, Grigorios Raptopoulos, Ioannis Anastopoulos, Dimitrios A. Giannakoudakis, Michael Arkas, Patrina Paraskevopoulou, Ioannis Pashalidis

**Affiliations:** 1Radioanalytical and Environmental Chemistry Group, Department of Chemistry, University of Cyprus, P.O. Box 20537, Nicosia CY-1678, Cyprus; 2Inorganic Chemistry Laboratory, Department of Chemistry, National and Kapodistrian University of Athens, Panepistimiopolis Zografou, 15771 Athens, Greece; 3Department of Agriculture, University of Ioannina, UoI Kostakii Campus, 47100 Arta, Greece; 4Department Chemistry, Aristotle University of Thessaloniki, 54124 Thessaloniki, Greece; 5Demokritos National Centre for Scientific Research, Institute of Nanoscience and Nanotechnology, 15771 Athens, Greece

**Keywords:** aerogels, environmental remediation, uranium adsorption, environmental water decontamination, adsorption thermodynamics and kinetics, extraordinary adsorption capacity, *q*_max_ values, competitive ions, material recycling, uranium recovery

## Abstract

Aerogels are a class of lightweight, nanoporous, and nanostructured materials with diverse chemical compositions and a huge potential for applications in a broad spectrum of fields. This has led the IUPAC to include them in the top ten emerging technologies in chemistry for 2022. This review provides an overview of aerogel-based adsorbents that have been used for the removal and recovery of uranium from aqueous environments, as well as an insight into the physicochemical parameters affecting the adsorption efficiency and mechanism. Uranium removal is of particular interest regarding uranium analysis and recovery, to cover the present and future uranium needs for nuclear power energy production. Among the methods used, such as ion exchange, precipitation, and solvent extraction, adsorption-based technologies are very attractive due to their easy and low-cost implementation, as well as the wide spectrum of adsorbents available. Aerogel-based adsorbents present an extraordinary sorption capacity for hexavalent uranium that can be as high as 8.8 mol kg^–1^ (2088 g kg^–1^). The adsorption data generally follow the *Langmuir* isotherm model, and the kinetic data are in most cases better described by the pseudo-second-order kinetic model. An evaluation of the thermodynamic data reveals that the adsorption is generally an endothermic, entropy-driven process (Δ*H*^0^, Δ*S*^0^ > 0). Spectroscopic studies (e.g., FTIR and XPS) indicate that the adsorption is based on the formation of inner-sphere complexes between surface active moieties and the uranyl cation. Regeneration and uranium recovery by acidification and complexation using carbonate or chelating ligands (e.g., EDTA) have been found to be successful. The application of aerogel-based adsorbents to uranium removal from industrial processes and uranium-contaminated waste waters was also successful, assuming that these materials could be very attractive as adsorbents in water treatment and uranium recovery technologies. However, the selectivity of the studied materials towards hexavalent uranium is limited, suggesting further developments of aerogel materials that could be modified by surface derivatization with chelating agents (e.g., salophen and iminodiacetate) presenting high selectivity for uranyl moieties.

## 1. Introduction

Rapid industrial development and enormous technological progress in the last few decades resulted in the accumulation of organic chemicals (e.g., dyes, pesticides, and pharmaceuticals etc.), heavy/toxic metals, metalloids, and radionuclides, mainly in waste form, polluting the environment and threatening living organisms [1,2,3,4,5]. Pollution related to (radio)toxic metals and metalloids is of particular interest because of their persistence and complex environmental chemistry. Metal/metalloid ions can enter water bodies after their dissolution and extraction from solid matrices or the deposition of airborne particles. Then, they can go into the biosphere, including into larger organisms, mainly through the food chain [6,7].

Increased amounts of (radio)toxic metals and particularly uranium have accumulated in the environment, mainly through anthropogenic activities related to mining and ore beneficiation, energy production (e.g., nuclear power), fertilizer production, and the use of depleted uranium in armor-piercing ammunition and tank armor. Uranium, an actinide element, is of particular interest not only because of its chemical toxicity, but also because it is a radioactive element with its most abundant isotope having a very long half-life (*t*_1/2_ = 4.5 × 10^9^ years) and emitting high energy alpha-particles (~4.5 MeV) [8,9]. The aqueous chemistry of uranium can be very complex, since it can undergo various chemical transformations, such as reduction/oxidation, hydrolysis, complexation, colloid formation, sorption, and precipitation, forming a wide spectrum of chemical species that each behave differently in the aquatic systems [9,10,11].

Regarding the removal of uranium from industrial process solutions and contaminated waters, a large spectrum of treatment technologies (e.g., ion exchange, precipitation, solvent extraction, and adsorption) have been investigated. However, among them, adsorption is the most attractive route. Adsorption is a chemical process that includes a solid phase (adsorbent) and a liquid phase, which contains the soluble species to be adsorbed (adsorbate). The adsorption of uranium, which basically exists in aqueous solutions in its hexavalent form (U(VI)), occurs via pure electrostatic attraction between the oppositely charged surface of the adsorbate and/or via direct binding between surface active groups (e.g., –OH and –COOH) and U(VI) [12,13,14,15]. The latter results in the formation of inner-sphere complexes [13,14], whereas the former results mainly in the formation of outer-sphere complexes [16]. Studies on uranium sorption are both fundamental and necessary regarding the chemical behavior and mobility of this (radio)toxic element in the geosphere, the decontamination of waters, and the recovery of this precious metal from industrial processes and wastewater [1,17].

In recent years, investigations have been focused on the development and production of very effective and selective adsorbents. Such adsorbent materials include, but are not limited to, inorganic solids (metal oxides and minerals) [18], biomass by-products [12,13,14,15,19], composite materials [20,21,22,23], polymers [24,25], dendrimers [26], MOFs [27,28,29], carbon-based materials [30], hybrid materials [31,32,33], and biopolymer-based materials [34,35,36]. Recently, and more intensively in the last decade, aerogels of various chemical compositions and nanostructures have been tested for uranium adsorption and recovery from wastewater and seawater, sometimes with impressive performances [37,38,39,40,41,42,43,44,45,46,47,48,49,50,51,52,53,54,55,56,57,58,59,60,61,62,63,64,65,66,67,68,69,70,71]. Aerogels have also been used for the photocatalytic conversion of soluble uranium species to insoluble nanoparticles that can float on water [72] and as hosts for reagents that leach out and cause the precipitation of uranium from water [73].

This review presents the recent progress in the development and use of aerogel materials for uranium uptake and recovery from aquatic environments. Knowledge of the chemical behavior and speciation of hexavalent uranium at certain experimental conditions is of fundamental importance to better understand and describe the adsorption process. In addition, an insight into the physicochemical parameters affecting the adsorption efficiency and mechanism will enable the design and development of efficient and selective adsorbents.

### 1.1. Aerogels

Aerogels have been defined as solid colloidal or polymeric networks expanded throughout their entire volume by a gas [74,75]. In practice, aerogels are nanostructured, ultra-lightweight materials [76], consisting mostly of empty space (>80% *v*/*v*). They are prepared via sol–gel processes that yield gels, which are subsequently dried by turning the pore-filling solvent into a supercritical fluid and releasing it as a gas (Figure 1). The specific drying process is the key to avoiding significant volume reduction or network compaction during the transition from gels to aerogels, and therefore, is still the most widely used, although in some cases, materials with aerogel properties have been obtained after sub-critical drying, freeze-drying (cryogels), or even drying under ambient conditions [76].

S. S. Kistler was the first to prepare silica aerogels, the most well-known type of aerogel, in the 1930s, along with other metal oxide aerogels as well as organic aerogels [77,78,79,80]. Kistler’s first silica aerogels were commercialized through the Monsanto Chemical Company [81]. Later, in 1966 a new method, using alkoxides as aerogels precursors, was reported by J. B. Peri [82]. From the very first publication by Kistler [77], it was clear that this class of lightweight, nanoporous, and nanostructured materials is not limited to a certain chemical composition, not even to a certain class of chemical compounds, rather it can include several materials, ranging from inorganic to organic and from synthetic to natural polymers. Indeed, nowadays aerogels are a huge family of materials, including inorganic oxides, chalcogenides, metals, ceramics, natural and synthetic organic polymers, and carbons [76,83,84]. In fact, there are no chemical compounds that could not be made in an aerogel form [85]. Aerogels can be prepared in any desirable form factor, including mostly monoliths, but also blankets [86], fibers [87], films [88,89], and millimeter-sized beads or fine powders [90,91,92,93].

In addition to the chemical composition, the size and shape of the pores affect the properties of an aerogel, as is the case for all porous materials. Most aerogels are mesoporous materials with pore sizes in the 2–50 nm range. The solid network consists of primary particles that aggregate to form fractal porous secondary particles, eventually agglomerating to a “pearl-necklace” structure. The finely structured porous skeletal framework together with the small-sized pores provides aerogels with unique properties, among which are high surface areas, low thermal conductivities, low dielectric constants, and high acoustic attenuation [76]. Interestingly, the nanostructure of the aerogels can be designed and tuned by choosing specific monomers [94,95,96,97,98] or by modifying the synthetic procedure [99,100,101,102,103,104,105,106].

Based on the above, it is obvious that aerogels are extremely versatile and promising materials for a wide range of technological areas. Indeed, IUPAC has recently announced aerogels in the 2022 top ten emerging technologies in chemistry [107]. The areas of application include, but are not limited to, thermal (their flagship application) [86,108,109,110,111,112,113,114,115] and acoustic insulation [109,110,115], space applications [109,116], transparent materials [117,118,119] energy storage [100,108,120,121], dielectrics [109], gas and humidity adsorption [109,113,122,123,124,125], sensors [115,122], actuators [126,127,128], catalysis [100,113,122,124,129,130,131,132,133], biomedicine [134,135], the food industry [113,136], and environmental remediation [101,134,137,138].

Indeed, the potential of aerogels in the field of environmental remediation is shown by the dramatic increase in the number of publications (Figure 2a), especially in the last decade, and by the launching in 2019 of a COST Action entitled “Advanced Engineering and Research of aeroGels for Environment and Life Sciences” [139]. Relevant to this review, the number of publications on utilizing aerogels for uranium uptake has also been increasing rapidly over the last decade (Figure 2b).

### 1.2. Uranium

Uranium is a natural element and a member of the actinide series (5*f* elements). It has an atomic number of 92 and an atomic weight of 238.02891 g mol^–1^. Uranium is a relatively abundant element at a mean concentration of 2.4 ppm in the earth’s crust and about 3.3 ppb in the oceans. Natural uranium is a mixture of three isotopes U-238, U-235, and U-234, which is a daughter nuclide of U-238, with a relatively short half-life and is hence, more or less in radioactive equilibrium with its parent nuclide U-238. All uranium isotopes are alpha-particle-emitting radionuclides, but generally, uranium is a relatively weak radioactive element. However, uranium is a heavy element and is hence, chemotoxic above certain levels, which are far below the levels of its radiotoxicity [8].

Uranium in nature can exist in five different oxidation states from +2 to +6. However, +4 and +6 are the most abundant oxidation states, and +6 is the predominant oxidation state in aqueous solutions under ambient conditions. In hexavalent oxidation, state uranium exists in the form of the uranyl cation (UO_2_^2+^) and is easily hydrolyzed (pH > 4) in aqueous solutions. The formation of the polynuclear species is favored at an increased U(VI) concentration (>10^–5^ M). Moreover, under ambient conditions and in the presence of carbonate cations, the U(VI)-carbonato species govern the U(VI) chemistry in aqueous solutions and in the near neutral and alkaline pH region [10,11,140]. Figure 3 shows the solubility curve of UO_2_(OH)_2_, which is the solubility limiting solid phase of U(VI) under ambient conditions in aqueous solutions and the corresponding species distribution diagram, which includes only mononuclear U(VI) species. In order to denote the impact of carbonate complexation, Figure 3 also includes the solubility curve of UO_2_(OH)_2_ assuming only hydrolysis. The calculation of both the solubility curves and the species distribution has been performed using the solubility product of UO_2_(OH)_2_ and the formation constants of the hydrolysis species and the carbonate complexes that are available in the literature [10,11].

## 2. Uranium Sorption by Aerogels

As stated before and shown in Figure 2b, in recent years a relatively large number of studies have been published regarding the sorption of U(VI) by aerogels and particularly, the effect of several parameters (e.g., pH, initial U(VI) concentration, ionic strength, contact time, temperature, adsorbent dosage, and presence of competing species) that affect U(VI) sorption. In addition, some of these studies include the possible recovery of uranium and reusability of the aerogel adsorbent, as well as the employment of spectroscopic techniques in order to identify the surface species formed after the adsorption and get insight into the sorption mechanism. Table 1 summarizes the experimental parameters and thermodynamic data evaluated from the published studies on uranium adsorption by several aerogel materials [37,38,39,40,41,42,43,44,45,46,47,48,49,50,51,52,53,54,55,56,57,58,59,60,61,62,63,64,65,66,67,68,69,70].

Before we proceed to the analysis of the parameters that affect U(VI) sorption, we need to answer one basic question. How important is the nanostructure of aerogels for the specific application sorption? To answer this question, we will use two examples. First, we will compare the maximum sorption (*q*_max_) reported for calcium alginate xerogels (appr. 9 g kg^–1^) [36] and calcium alginate aerogels (388 g kg^–1^; Table 1) [39] at pH 3. Since the chemical composition of the adsorbent and the reaction conditions are the same, the huge difference in *q*_max_ must be credited to the nanostructured porous calcium alginate aerogel. Second, we will compare polyurea-crosslinked calcium alginate (X-alginate) aerogels with an aliphatic [141,142,143] or an aromatic [144] polyurea. In both cases, the alginate network is covered by polyurea and the materials have similar material properties and morphologies; however, there is a critical difference in the nanostructure: the aliphatic polyurea forms a compact layer that covers the alginate primary nanoparticles, while the aromatic polyurea has a more rigid and randomly oriented polymer structure that partially fills the pores within the secondary particles [95]. Because of this difference, these two materials behave differently versus the sorption of heavy metal ions. X-alginate aerogels with the aromatic polyurea can efficiently uptake Pb(II) [145], U(VI) [39], Eu(III) and Th(IV) [146], while X-alginate aerogels with the aliphatic polyurea are not so efficient. For example, for Pb(II), the *q*_max_ values are equal to 20.8 g kg^–1^ and 6.8 g kg^–1^, respectively [147].

### 2.1. pH Effect

pH is one of the most important parameters regarding sorption, since it affects both the speciation of the element in the solution and the degree of dissociation of the functional surface groups. Under ambient conditions, U(VI) speciation in aqueous solutions includes hydrolysis species (e.g., UO_2_^2+^, UO_2_OH^+^, UO_2_(OH)_2_, (UO_2_)_2_(OH)_2_^2+^, (UO_2_)_2_OH_3_^+^, (UO_2_)_3_(OH)_5_^+^, and (UO_2_)_4_(OH)_7_^+^) which predominate at pH < 6 and U(VI) carbonate complexes (UO_2_CO_3_, UO_2_(CO_3_)_2_^2−^, UO_2_(CO_3_)_3_^4−^), which govern the U(VI) chemistry at pH > 6. At pH < 4, the uranyl ion (UO_2_^2+^) dominates and determines the U(VI) chemistry in the solution [10].

U(VI) sorption by aerogels has been investigated in a wide pH range (3–11) and the sorption capacity (*q*_e_) has been determined for pH values ranging between 3 and 8, and the total U(VI) concentration is close to or even above the solubility limit of the predominant U(VI) solid phase under ambient conditions (e.g., UO_2_(OH)_2_; Figure 3). This is crucial, particularly in the near neutral pH range (5 < pH < 7) because UO_2_(OH)_2_ is very likely to be present, interfering with sorption and resulting in erroneous conclusions. In acidic (pH < 5) and alkaline (pH > 7) solutions, the solubility increases significantly due to acidic solid phase dissolution and carbonate complexation of U(VI). It is obvious that studies related to pH-effect should be performed at total U(VI) concentrations below the solubility limits to avoid artifacts associated with solid-phase precipitation. In addition, the formation of hydrolysis and polynuclear species for pH > 4, as well as carbonate complexes for pH > 6, is expected to result in various sorption interactions which may also differ from one aerogel type to another. Therefore, a large variation is observed regarding the pH values at which the maximum sorption capacity (*q*_max_) occurs, even for similar types of aerogel materials. In some cases, there is a range of several pH units in which the materials present their *q*_max_, while in other cases, the *q*_max_ is observed at a certain pH value.

In order to obtain more reliable and comparable results, we suggest performing the sorption experiments associated with the evaluation of *q*_max_ at a certain value in the acidic pH range, such as pH 3 or pH 4; the former is preferable for aerogel materials, in order to carry out experiments with the total U(VI) concentration above the mmol range, which is needed to reach the plateau of the isothermal curves and accurately evaluate *q*_e_ values [39]. Indeed, the first three aerogels in terms of sorption capacity (Table 1) show maximum capacity in acidic solutions (pH 3–4), in which range UO_2_^2+^ is the only species present [37,38,39,40].

At the molecular level, the optimum pH is associated, in the case of inorganic oxides (e.g., Al_2_O_3_ and TiO_2_), with the point of zero charge (pzc) of the materials and the interaction of the negatively charged surface with cationic U(VI) species at pH < 8 [10,11]. Below the pzc of the respective oxide, the adsorbent surface is positively charged due to surface protonation (=S–OH + H^+^_(aq)_ ⇌ =S–OH_2_^+^; S: surface) and the sorption efficiency is low due to the electrostatic repulsion between the adsorbent surface and the cationic U(VI) species (e.g., UO_2_^2+^, UO_2_OH^+^, and (UO_2_)_2_(OH)_2_^2+^). At pH values above the pzc, the surface hydroxyl groups are deprotonated (=S–OH + H_2_O ⇌ =S–O^–^ + H_3_O^+^) and the surface attracts the positively charged U(VI) resulting in enhanced sorption. However, above pH 6 and under ambient conditions, the carbonate concentration in a solution increases progressively, resulting in the formation of very stable U(VI)-carbonato complexes, which stabilize U(VI) in solution and compete for U(VI) surface complexation [10,140,148].

Similarly, in the case of carbon-based aerogels, the optimum pH lies in the weak acidic pH range (pH~5) and is determined by the competing reactions: (a) the protonation of surface active groups and (b) the formation of U(VI) hydroxo- and carbonate complexes, which stabilize U(VI) in solution [10,140]. Under ambient conditions, the latter complexes are expected to govern the U(VI) in the solution, and have to be considered in associated species distribution diagrams. Adsorption on modified carbons has been observed via the interaction of the metal species with sulfur (e.g., –SO_3_^–^) [68] or phosphorous (e.g., –O–PO_3_) functionalities [67] at pH values 5–5.5.

### 2.2. Sorption Kinetics and Kinetic Modelling

In large-scale and industrial applications, fast and efficient sorption can reduce production costs and at the same time, accelerate production efficiency. The adsorption kinetics describe the rate of adsorbate uptake on the adsorbent, which determines the adsorption equilibrium time. Therefore, investigations on sorption kinetics and kinetic data modelling are of fundamental importance. Most studies use the pseudo-first-order Equation (1) and pseudo-second-order Equation (2) adsorption kinetic models to analyze the experimental data.
(1)$$q_{t}=q_{e}(1-\;e^{-k_{1}t})$$
(2)$$q_{t}=\frac{k_{2}q_{e}{}^{2}t}{1+k_{2}q_{e}t}$$

Generally, due to the large number of active sites and functional groups on the surface of the aerogel materials, the U(VI) adsorption capacity increases rapidly with time. Generally, in an initial relatively fast step, most of the U(VI) is adsorbed, and sorption continues until the systems reach equilibrium. The second step is characterized by a lower adsorption rate, which is attributed to the gradual occupation of surface active sites and the decrease of the U(VI) concentration in a solution [63]. With the exception of alumina-based aerogels [44], which need about 300 min to reach a steady-state, the sorption on purely inorganic aerogels reaches its maximum values within a few minutes [37,43,47]. Graphene-based aerogels reach equilibrium conditions mainly after ~50 min [62] and biomass-derived carbon aerogels need from 100 up to 1500 min to reach equilibrium [63], with the exception of polyurea-crosslinked calcium alginate (X-alginate) and calcium alginate aerogels, which reach equilibrium within a few minutes [39]. Equally, fast adsorption was also observed for the corresponding polyurea aerogels [39].

Based on the linear correlation coefficients (*R*^2^) and the calculated maximum adsorption capacity (*q*_max,cal_), which in the case of the pseudo-second-order kinetic model, are close to unity and the experimental maximum adsorption capacity (*q*_max,exp_), respectively, sorption data are better described by the pseudo-second-order kinetic model, which could be in agreement with chemisorption. However, this is an over-simplification of the parameters that affect the adsorption; for example, the diffusion into the porous material should always be taken into consideration [149]. Recent research supports the application of the non-linear form of the pseudo models for analyzing the adsorption kinetics, otherwise erroneous conclusions may be derived [150].

### 2.3. Sorption Isotherms

The effect of the initial concentration on the sorption efficiency at a constant temperature is of particular interest because it enables the fitting of the experimental data and the evaluation of the maximum adsorption values by applying simple empirical models (usually, *Langmuir* and *Freundlich* adsorption isotherm models). Adsorption isotherms are essential for expressions of the theoretical maximum adsorption capacities and surface characteristics of the adsorbents, adsorption mechanism pathway optimization, and the productive design of the adsorption systems since they explain how model pollutants are interrelated with the materials of the adsorption process (adsorbents) [151]. Although in some cases, these models are used to evaluate the adsorption mechanism (e.g., chemisorption) [72], this approach is very vague and should be used only as an indication.

According to Table 1, the maximum adsorption values (*q*_max_) extend in a wide range between 13 g kg^–1^ and 2088 g kg^–1^, with the highest values being among the highest ever reported for uranium adsorption. For example, the recently reported values above 1800 g kg^–1^ are by far the highest found in the literature [37,38,39,40]. This is associated with the large surface area and hence the large number of active sites available for U(VI), binding on the aerogel materials. It must be noted that extremely high *q*_max_ values have been reported for inorganic (hydroxyapatite) [37,38], biopolymer-based (X-alginate) [39], and carbon-based aerogels [40]. The only higher value than the above (*q*_max_ = 3550 g kg^–1^) has been reported for graphene-based aerogels, which however, act not as adsorbents, but as hosts for reagents that leach out of the aerogel matrix and cause precipitation of uranium from water [73].

### 2.4. Effect of Solution Composition and Competing Ions

In order to simulate real-world conditions, several investigations have focused on the effect of co-existing ions, such as K^+^, Na^+^, Ca^2+^, Mg^2+^, CO_3_^2–^, PO_4_^3–^, SO_4_^2–^, Cl^–^, ClO_4_^–^, and NO_3_^–^, on the U(VI) sorption by aerogels [35,37,44,45,59,63]. According to the related studies and the corresponding data that are summarized in Figure 4, conservative cations (e.g., K^+^ and Na^+^) do not remarkably affect sorption efficiency, whereas in the presence of polyvalent metal ions (e.g., Ca^2+^, Zn^2+^, and Al^3+^) there is a significant decrease of the relative sorption efficiency [35,37,40,41,43,44,45,46,47,49,50,52,53,56,58,59,60,64,67,68]. This occurs because polyvalent metal cations can interact with surface moieties to form complexes, and therefore, compete with U(VI) by occupying surface binding sites (Figure 5 top). The absence of any effect in the case of the conservative cations is a clear indication that U(VI) binding by the aerogels surface is based on specific interactions, which result in inner-sphere surface complexes, not only because of the pure electrostatic interactions associated with the formation of outer-sphere complexes [12,13,14,15].

On the other hand, among the studied anions (Figure 4), only the presence of CO_3_^2–^ and PO_4_^2–^ in the solution seemed to significantly reduce the relative U(VI) removal, because both CO_3_^2–^ and PO_4_^2–^ form very stable complexes with UO_2_^2+^ (e.g., UO_2_CO_3_, UO_2_(CO_3_)_2_^2−^, UO_2_PO_4_^−^, UO_2_HPO_4_), which govern the U(VI) chemistry in the system [56,60,152]. The competitive interaction between the U(VI) and carbonate cations to form the U(VI)-tricarbonato complex, which stabilizes U(VI) in aqueous solutions, is schematically shown in Figure 5 (bottom).

Data obtained from these studies clearly show that the selectivity of the aerogel materials for uranium is limited, and that the sorption capacity decreases remarkably in more complex aqueous laboratory solutions [37,42,44,65,68] and natural waters (e.g., seawater [39,42,48]). Hence, specific surface modifications (e.g., surface derivatization with salophen [153]) are needed to substantially increase selectivity toward the U(VI), which will only be insignificantly affected by the presence of competing polyvalent metal ions, strong complexing anions (e.g., CO_3_^2–^), and acidic conditions (pH < 3).

### 2.5. Temperature Effect and Sorption Thermodynamics

Experiments related to the effect of temperature enable an evaluation of sorption thermodynamics and the calculation of the associated parameters (e.g., Δ*H*^0^, Δ*S*^0^, and Δ*G*^0^). The values of the thermodynamic parameters indicate whether sorption is an exothermic (Δ*H*^0^ < 0) or an endothermic (Δ*H*^0^ > 0), entropy-driven (Δ*S*^0^ > 0) process. The Gibbs free energy change Δ*G*^0^ (kJ mol^–1^) can be calculated from Equation (3) (van’t Hoff equation), and it is connected with the enthalpy change (Δ*H*^0^) and the entropy change (Δ*S*^0^) through Equation (4), where *R* is the gas constant (8.314 J mol^–1^ K^–1^), *T* is the absolute temperature (K) and $K_{e}^{0}$ is the thermodynamic equilibrium constant.
(3)$$\Delta G^{0}=-RT\ln K_{e}^{0}$$
(4)$$\Delta G^{0}=\Delta H^{0}-T\Delta S^{0}$$

With the exception of only one study [62], all other studies up to now have indicated that U(VI) sorption by aerogel materials is an endothermic, entropy-driven process that is favored with increasing temperatures [39,40,50,52,53,55,56,59,62,63,65,66,68,69]. Δ*H*^0^ values can be as high as 141 kJ mol^–1^ and Δ*S*^0^ values can be as high as 500 J K^–1^ mol^–1^ (Table 2). The increase of entropy, which is the main driving force for U(VI) sorption by aerogels, is ascribed to the release of water molecules from the hydrated U(VI) ionic species and the charged surface moieties upon the U(VI) surface complexation, as schematically indicated in Figure 6.

### 2.6. Uranium Recovery and Material Recycling

Material recycling and uranium recovery are of particular interest from an environmental and economic point of view. The former is related to the treatment of uranium-contaminated waters and uranium monitoring in the environment, and the latter is related to the recovery of precious and industrial metals to cover increasing demands and compensate for the decline of natural resources. Regeneration studies have been carried out using EDTA solutions as extractants because the EDTA has a strong ability to complex U(VI) without causing any damage to the adsorbent material [39]. Usually, five consecutive adsorption-desorption cycles are performed, and the recovery and regeneration efficiency are quantified to evaluate the process’ applicability. The related studies have shown that the recovery and regeneration efficiency were satisfactory and that the aerogel material remained almost intact [34,39,42,44,55,70]. The use of nitric acid solutions (0.5 M HNO_3_) can result in the deterioration of the aerogel porous structure during the adsorption–desorption process and subsequently, to the U(VI) recovery decline [65].

Despite the fact that carbonate solutions have been used as extractants for uranium recovery in other sorption studies [22] or often for the extraction of uranium from minerals/rocks, there are only a few studies [39] on the removal of U(VI) from aerogel materials making use of the high carbonate affinity for U(VI) to form very stable U(VI)-carbonato complexes [148], which is obvious from the solution composition and competing ions studies.

### 2.7. Effect of the Adsorbent Mass and Ionic Strength

Among the factors that affect the U(VI) sorption by aerogels and have been investigated is the adsorbent dose. Generally, increasing the adsorbent dose results in an increase in the removal capacity but a gradual decrease of the relative removal [43,63]. Increasing the adsorbent dose is associated with an increase in the available surface area and the number of active sites, and positively affects the sorption capacity, which reaches a maximum value and a plateau after a certain adsorbent amount. On the other hand, the relative sorption efficiency decreases because, usually, an association of the adsorbent particles can lead to a reduced relative adsorption efficiency.

Investigations on the effect of ionic strength on the adsorption could provide indications about the predominance of inner- or outer-sphere complex formations. Generally, when inner-sphere complexes are formed, which are characterized by a direct chemical bond between the surface functional groups of the adsorbent and the adsorbate, the sorption efficiency does not significantly depend on the ionic strength of the solution. On the other hand, if the adsorption is controlled mainly by outer-sphere complexes, which are based on pure electrostatic attraction and the ionic adsorbate retains its hydration sphere, the sorption efficiency is strongly affected by the ionic strength/salinity [152,154].

### 2.8. Spectroscopic Studies and Sorption Mechanism

Generally, the adsorption mechanism of U(VI) by aerogels is investigated by FTIR and XPS spectroscopy. Regarding FTIR, peak shifts and relative changes in peak intensities of surface-active groups (e.g., –COOH and –OH,) as well as the appearance of the characteristic uranyl (O=U=O) vibration in the spectrum after uranium sorption are employed to evaluate the adsorption mechanism. Interestingly, the peak shift of the uranyl moiety differs significantly from one type of aerogel to another, and the values of the peak maximum vary between 895 cm^–1^ and 918 cm^–1^ [38,39,44,53,59,62,64,66]. On the other hand, changes in the relative area and binding energy of carbon and oxygen peaks associated with surface active moieties after U(VI) adsorption are used as a clear indication for surface complex formation [41,44,49,59,66]. In addition, the peaks at 383 eV in the high-resolution XPS spectra associated with U_4*f*_ are used to indicate complexation with surface groups [62]. EDS spectroscopy has also been applied after the adsorption tests to confirm the presence of uranium in the adsorbent material [39,55].

The information obtained from FTIR, XPS, and EDS spectroscopic studies is very useful, and the associated data clearly indicates the surface-active groups that interact and form covalent bonds with the adsorbed uranium. However, a comprehensive description of the U(VI) binding on the aerogel surface at the molecular level is missing. In this context, EXAFS measurements along with theoretical calculations (e.g., DFT calculations [49]) would provide further insight into the adsorption mechanism. In addition to the spectroscopic methods, surface zeta potentials were used to point out the role of the surface charge with respect to U(VI) adsorption by aerogels materials [41].

Regarding the evaluation of the sorption mechanism, the spectroscopic measurements are of particular interest because the thermodynamic and kinetic data obtained from the sorption experiments are not specific for a single/defined reaction but correspond to an overall sorption reaction, which is the sum of the separate sorption reactions occurring at the aerogel surface. The number of different sorption reactions depends on the surface homogeneity, the different active groups available on the surface, and the U(VI) species that dominate in solution. The latter becomes significant for pH > 4 when hydrolysis and carbonate complexation are governing the solution chemistry of U(VI) [10,11,140].

### 2.9. Bulk Density of the Aerogel Material

Another property that must be taken into consideration when the practical applications and real-world conditions are targeted, is the bulk density (*ρ*_b_) of the aerogel material. Aerogels are famous for their extremely low bulk densities; however, in this particular case, this may not be an advantage. Indeed, our previous works [39,155] have shown that the calculation of the adsorption efficiency in g per liter (*q*_max(V)_) of the aerogel material (instead of g per kg (*q*_max_); Table 1) is very important and provides an estimation of the volume of the material needed for the uptake of a certain amount of uranium. For example, if we compare four aerogels from Table 1, i.e., polyurea-crosslinked alginate (X-alginate; *q*_max_ = 2023 g kg^–1^, *ρ*_b_ = 150 g L^–1^, *q*_max(V)_ = 303 g L^–1^) [39], Al_2_O_3_/MgO (*q*_max_ = 1046.9 g kg^–1^, *ρ*_b_ = 18.89 g L^–1^, *q*_max(V)_ = 19.8 g L^–1^) [41], calcium alginate (*q*_max_ = 388 g kg^–1^, *ρ*_b_ = 68 g L^–1^, *q*_max(V)_ = 26.4 g L^–1^) [39], and aromatic polyurea derived from TIPM (*q*_max_ = 305 g kg^–1^, *ρ*_b_ = 150 g L^–1^, *q*_max(V)_ = 45.8 g L^–1^) [39], it is obvious that X-alginate aerogels, with the highest density among the four aerogels, outperform the other three aerogels by far. That means that for the removal of 300 g of uranium, one would need 15.9 L of Al_2_O_3_/MgO, 11 L of calcium alginate, 6.6 L of polyurea, but only 1 L of X-alginate aerogels. A detailed presentation of *q*_max(V)_ for all aerogel materials for which bulk densities are reported has been published in reference [39].

## 3. Conclusions and Future Studies

Over the last decade, the number of studies concerning uranium sorption using aerogels has dramatically increased. This is because of the steadily increasing interest in the production of efficient adsorbent materials for radionuclide removal and recovery from contaminated waters. Thus, captured uranium is planned to cover the future demands of the nuclear power industry.

Generally, studies are focused on the effect of different physicochemical parameters (e.g., pH, initial U(VI) concentration, ionic strength, temperature, and contact time) on the adsorption efficiency. According to those studies, the pH plays a key role because it governs both the U(VI) speciation and surface species dissociation and charge.

The isothermal data obtained from experiments related to the effect of metal ion concentration are best fitted by the *Langmuir* adsorption isotherm model, resulting in supreme sorption capacity values (in some cases even above 2000 g of uranium per kg of aerogel) that are the highest ever reported for uranium. The experimental kinetic data indicate fast sorption kinetics (equilibrium is reached in a few minutes to a few hours), and in most cases they are best fitted by the pseudo-second-order kinetic model. Both the *Langmuir* isotherm and the pseudo-second-order kinetic models are indicative of chemisorption. Other parameters/phenomena, however, such as the diffusion within the porous matrix of aerogels, the adsorbent dose, or the effect of the ionic strength, cannot be ignored. In addition, the associated thermodynamic data generally reveal an endothermic, entropy-driven sorption mechanism, indicating the formation of inner-sphere surface complexes. The formation of inner-sphere complexes between U(VI) and the active groups on the aerogel surfaces is supported mainly by FTIR and XPS data.

Several aerogel materials show an excellent performance regarding their reuse, even after several adsorption-desorption cycles, with a relatively high uranium recovery. However, the presence of multivalent metal cations (e.g., Ca^2+^ or Al^3+^) and complexing species (e.g., CO_3_^2−^ or PO_4_^3−^) strongly affects the sorption efficiency of the adsorbents toward uranium, because of competitive sorption and the formation of stable solution complexes, respectively, indicating a limited selectivity of the studied aerogels toward uranium (U(VI).

Based on this review, future studies should focus on the preparation of aerogel materials with specific surface groups possessing a high affinity and selectivity toward uranium and other precious and industrial metals/metalloids. This is of particular interest because, besides sorption affinity and capacity, selectivity is a key factor affecting the recovery of the desired metal from multicomponent and complex industrial processes and waste waters. In addition, in radiopharmaceutical applications, it is of cardinal importance to selectively separate and recover the radionuclide of interest from reaction solutions containing other radionuclides and undesirable by-products.

The most efficient and selective aerogel materials should be tested not only in laboratory settings, but also at pilot and industrial scales in order to attract broader interest and find applications in large-scale/industrial processes. Moreover, data obtained from EXAFS (Extended X-ray absorption fine structure spectroscopy) and TRLFS (Time-resolved laser fluorescence spectroscopy) studies would allow a better understanding of the interaction between U(VI) species and the aerogel surface and describe the mechanism at the molecular level. This is of fundamental importance for the design and development of more effective and selective adsorbents.

## Figures and Tables

**Figure 1 nanomaterials-13-00363-f001:**
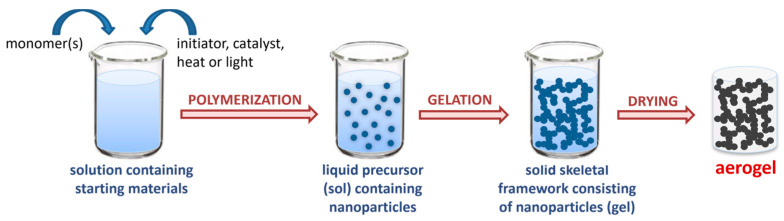
Schematic representation of the aerogel formation main steps.

**Figure 2 nanomaterials-13-00363-f002:**
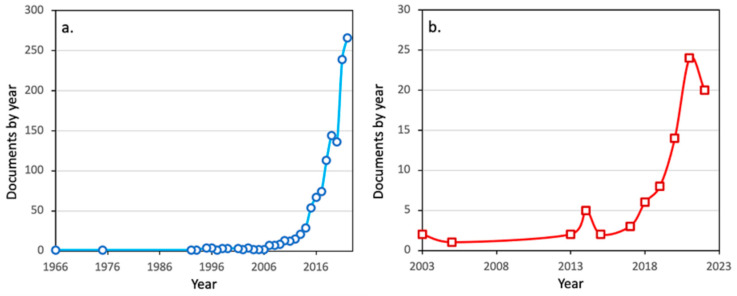
The number of publications on aerogels for environmental applications (**a**) and aerogels for uranium uptake and recovery (**b**). Source: Scopus (1 December 2022).

**Figure 3 nanomaterials-13-00363-f003:**
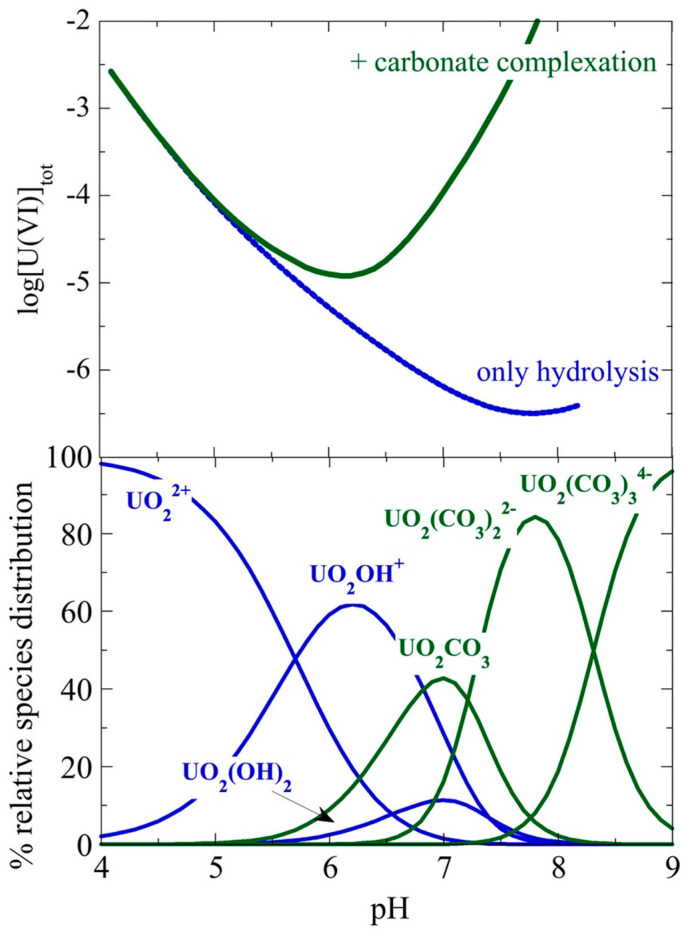
Top: solubility curve of UO_2_(OH)_2_ assuming only hydrolysis and under ambient conditions of hydrolysis and carbonate complexation. Bottom: speciation diagram including only U(VI) mononuclear species at various pH values.

**Figure 4 nanomaterials-13-00363-f004:**
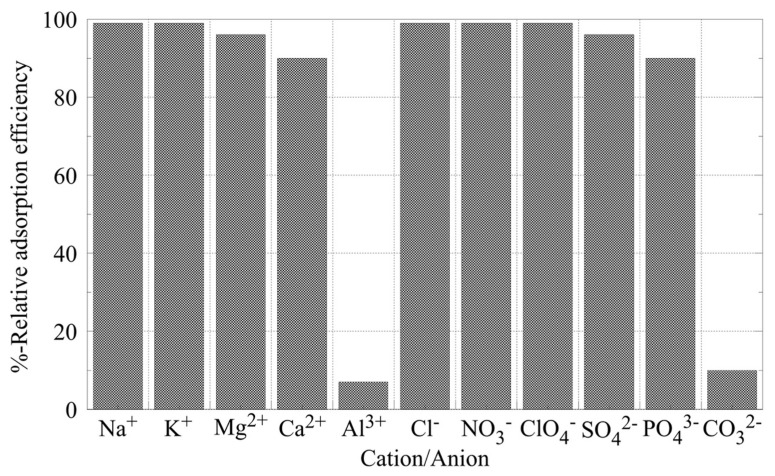
The effect of co-existing ions on the sorption of U(VI) by inorganic aerogels [42,44].

**Figure 5 nanomaterials-13-00363-f005:**
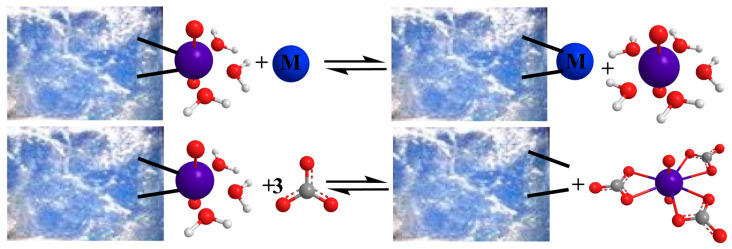
Schematic illustration of the competition reaction between U(VI) and competing cations (M^z+^) regarding the sorption on aerogel surfaces (**top**) and the stabilization of U(VI) in solution in the presence of carbonate ions by complex formation, which competes with surface complexation and sorption (**bottom**). Charges are omitted for simplicity.

**Figure 6 nanomaterials-13-00363-f006:**
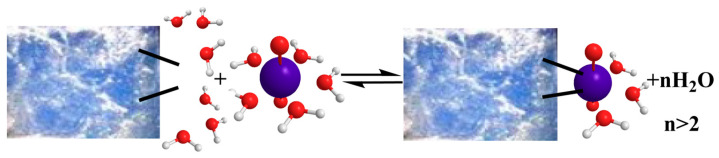
Schematic illustration of the water molecule released upon adsorption, associated with the entropy increase leading to an entropy-driven adsorption process. Charges are omitted for simplicity.

**Table 1 nanomaterials-13-00363-t001:** Selected experimental data for the best-fitted kinetics/isotherm models for the sorption of uranium by aerogel materials.

Aerogel Material	pH	Temp. (K)	[U(VI)]_o/max_ (mol L^–1^)	Best-Fitted Isotherm/Kinetic Model	*q*_max_ (g kg^–1^)	Competition/Recycling/Recovery	Data Related to U(VI) Adsorption	Ref.
Hydroxyapatite (templated with konjac gum)	4	298	4.2 × 10^–4^	*Langmuir*, PSO	2088	anion/cation competition reuse for five cycles	FTIR, XPS, and mechanistic studies	[37,38]
Polyurea-crosslinked alginate (X-alginate)	3	298	4.2 × 10^–5^	*Langmuir*	2023	natural waters, seawater, modelling, wastewater, and recycling	FTIR and EDS	[39]
Reduced graphene oxide/ZIF-67 ^a^	4	298	1.05 × 10^–3^	*Langmuir*, PSO	1888	cation competition, reuse for five cycles	FTIR and XPS	[40]
Al_2_O_3_/MgO	6	298	4.2 × 10^–5^	*Langmuir*, PSO	1047	cation competition, reuse for five cycles	XPS and mechanistic studies	[41]
MOF/black phosphorus quantum dots on cellulose ^b^	7	303		*Langmuir*, PSO	858	seawater and recycling	XPS and mechanistic studies	[42]
Pr_2_O_3_	7	298	4.2 × 10^–5^	*Langmuir*, PSO	841	cation competition and reuse for five cycles	FTIR and XPS	[43]
Al_2_O_3_ (templated with chitosan)	7	298	4.2 × 10^–5^	*Langmuir*, PSO	814	anion/cation competition and reuse for five cycles	FTIR, XPS, and mechanistic studies	[44,45]
Al_2_O_3_ (templated with polyethylene glycol)	7	298	4.2 × 10^–5^	*Langmuir*, PSO	737	anion/cation competition and reuse for five cycles	FTIR, XPS, and mechanistic studies	[44]
Amidoxime-functionalized *β*-cyclodextrin/graphene	6	298	8.4 × 10^–4^	*Langmuir*	654	cation competition and reuse for ten cycles	FTIR and XPS	[46]
TiO_2_	5	298	4.2 × 10^–5^	*Langmuir*, PSO	638	cation competition and reuse for five cycles	FTIR and XPS	[47]
Al_2_O_3_ (prepared with thiourea)	7	298	4.2 × 10^–5^	*Langmuir*, PSO	634	seawater	FTIR and XPS	[48]
Al_2_O_3_ (templated with dopamine)	7	298	4.2 × 10^–5^	*Langmuir*, PSO	592	anion/cation competition and reuse for five cycles	FTIR, XPS, and mechanistic studies	[44]
Poly(amidoxime)/graphene oxide nanoribbons	4.5	298	5.0 × 10^–4^	*Langmuir*	589	cation competition and reuse for five cycles	XPS, mechanistic studies, and DFT modelling	[49]
Nd_2_O_3_	7	-	4.2 × 10^–5^	*Langmuir*, PSO	587	cation competition and reuse for five cycles	FTIR and XPS	[43]
Bacterial cellulose@ZIF-8 carbon ^c^	3	308	8.4 × 10^–5^	*Langmuir*, PSO	535	cation competition and reuse for five cycles	FTIR and XPS	[50]
Calcium alginate/MgAlFe layered double hydroxides	5	298	8.4 × 10^–5^	*Langmuir*, PSO	532	-	FTIR and XPS	[51]
CeO_2_	7	298	4.2 × 10^–5^	*Langmuir*, PSO	482	cation competition and reuse for five cycles	FTIR and XPS	[43]
Nanocellulose	5	298	4.2 × 10^–5^	*Langmuir*, PSO	441	cation competition and reuse for five cycles	FTIR and XPS	[52]
Chitosan/aluminum sludge composite	4	308	3 × 10^–3^	*Langmuir*, PSO	435	cation competition and reuse for five cycles	XPS and mechanistic studies	[53]
Graphene oxide nanoribbon	4.5	298	2.52 × 10^–4^	*Langmuir*, PSO	431	-		[54]
Calcium alginate	3	298	4.2 × 10^–5^	*Langmuir*	388	-		[39]
Iron-polyaniline-graphene composite	5.5	318	4.2 × 10^–5^	*Langmuir*, PSO	350	reuse for five cycles	FTIR, XPS, EDS, and mechanistic studies	[55]
Chitosan/carboxylated carbon nanotube composite	5	318	5.04 × 10^–4^	*Langmuir*, PSO	341	cation competition	FTIR and XPS	[56]
Bayberry tannin/graphene composite	5	298	-	-	330	-		[57]
Reduced graphene oxide/g-C_3_N_4_ quantum dots/ZIF-67 composite carbon ^a^	3	328	8.4 × 10^–5^	*Langmuir*, PSO	316	cation competition and reuse for five cycles	FTIR and XPS	[58]
Aromatic polyurea derived from TIPM ^d^	3	298	4.2 × 10^–5^	*Langmuir*	305	-		[39]
Fungus hypha/graphene oxide	5	293	5.04 × 10^–4^	*Langmuir*	288	cation competition and reuse for six cycles	XPS	[59]
Aramid/polyamidoxime	6	298	4.2 × 10^–4^	*Langmuir*, PSO	262	cation competition and reuse for five cycles		[60]
Pr_2_O_3_	8	298	4.2 × 10^–5^	-	252	-		[61]
Graphene	4	298	1.02 × 10^–4^	*Langmuir*, PSO	239	reuse for four cycles	XPS	[62]
Carbon/Fe_3_O_4_	6	303	2.1 × 10^–4^	*Langmuir*, PSO	230	anion/cation competition and reuse for five cycles	FTIR and XPS	[63]
Melamine-formaldehyde/alginate	4	298	4.2 × 10^–4^	*Langmuir*	211	cation competition		[64]
Polydopamine- functionalized attapulgite/chitosan	6	-	2.1 × 10^–4^	*Langmuir*, PSO	175	reuse for six cycles	FTIR and XPS	[65]
*p*-Phthalaldehyde/3,5-diaminobenzoic acid-crosslinked chitosan	6	308	4.2 × 10^–5^	*Langmuir*, PSO	160	-	XPS	[66]
Phosphorylated carbon	5.5	298	4.2 × 10^–5^	*Langmuir*, PSO	150	cation competition and reuse for five cycles	FTIR and XPS	[67]
Sulfonated graphene	5	298	4.2 × 10^–5^	*Langmuir*, PSO	148	cation competition and reuse for five cycles	FTIR and XPS	[68]
Graphene oxide/carbon nanotubes	5	298	2.1 × 10^–4^	*Langmuir*, PSO	100	-		[69]
Graphene/Ag nanoparticles	5–6	298	8.4 × 10^–5^	*Langmuir*	13	-		[70]

^a^ ZIF-67: Co-based zeolitic imidazole framework. ^b^ Under simulated sunlight irradiation. MOF: UiO-66-NH_2_ (Zr-based metal-organic framework). ^c^ ZIF-8: Zn-based zeolitic imidazole framework. ^d^ Aromatic polyurea derived from Desmodur RE (27% w*/w* triphenylmethane-4,4′,4″-triisocyanate (TIPM) solution in ethyl acetate) from Covestro AG.

**Table 2 nanomaterials-13-00363-t002:** Literature values of Δ*H*^0^ and Δ*S*^0^ related to the U(VI) sorption by different aerogel materials.

Aerogel Material	Δ*H*^0^ (kJ mol^–1^)	Δ*S*^0^ (J K^–1^ mol^–1^)	Ref.
Reduced graphene oxide/ZIF-67 ^a^	11.7	120	[40]
Bacterial cellulose@ZIF-8 carbon ^b^	113.73	382.4	[50]
Nanocellulose	10.80	71.33	[52]
Chitosan/aluminum sludge composite	6.5	77	[53]
Iron-polyaniline-graphene composite	60.74	-	[55]
Chitosan/carboxylated carbon nanotube composite	21.96	157.3	[56]
Fungus hypha/graphene oxide	9.31	51.55	[59]
Graphene	−47.94	−73.03	[62]
Carbon/Fe_3_O_4_	141.4	500.2	[63]
Polydopamine- functionalized attapulgite/chitosan	5.45	50.24	[65]
*p*-Phthalaldehyde/3,5-diaminobenzoic acid-crosslinked chitosan	2.147	58.288	[66]
Sulfonated graphene	4.3	89.9	[68]
Graphene oxide/carbon nanotubes	8.146	91.43	[69]
Polyurea-crosslinked alginate (X-alginate)	>0	>0	[39]
Reduced graphene oxide/g-C_3_N_4_ quantum dots/ZIF-67 composite carbon ^a^	>0	>0	[58]

**^a^** ZIF-67: Co-based zeolitic imidazole framework. ^b^ ZIF-8: Zn-based zeolitic imidazole framework.

## Data Availability

Not applicable.

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
