# Peer review of "Uranium Removal from Aqueous Solutions by Aerogel-Based Adsorbents—A Critical Review"

_nanomaterials, 2023, doi:10.3390/nano13020363_

Round 1
Reviewer 1 Report
The manuscript reviews Uranium Removal from Aqueous Solutions by Aerogel-Based Adsorbents. There are some improvement before its publication.
1、 Abstract: the author should add the Uranium Removal background and the existing removal methods. And also should explain the purpose and significance of this review.
2、 Introduction: the last paragraph is too sudden, and some cohesive statements should be added.
3、 Figure 1 is too simple, and the author should enrich it. Figure 5 and Figure 6 should be merged.
4、 Conclusions: it is a repeated expression with the body content, and it should be reorganized.
5、 The language expression should be polished.
Author Response
Reviewer 1
1、 Abstract: the author should add the Uranium Removal background and the existing removal methods. And also should explain the purpose and significance of this review.
our response: the abstract has been revised and now reads as follows:
Aerogels are a class of lightweight, nanoporous and nanostructured materials with diverse chemical compositions and huge potential for applications in a broad spectrum of fields that have led IUPAC to include them in the top ten emerging technologies in Chemistry for 2022. This review provides an overview of aerogel-based adsorbents that have been used for the removal and recovery of uranium from aqueous environments, as well as an insight into the physicochemical parameters affecting adsorption efficiency and mechanism. Uranium removal is of particular interest regarding uranium analysis and recovery to cover present and future uranium needs for nuclear power energy production. Among the methods used, such as ion exchange, precipitation and solvent extraction, adsorption-based technologies are very attractive due to their easy and low-cost implementation, as well as the wide spectrum of adsorbents available. Aerogel-based adsorbents present extraordinary sorption capacity for hexavalent uranium that can be as high as 8.8 mol kg–1 (2088 g kg–1). The adsorption data generally follow the Langmuir isotherm model, and the kinetic data are in most cases better described by the pseudo-second-order kinetic model. Evaluation of the thermodynamic data reveals that the adsorption is generally an endothermic, entropy-driven process (ΔHo, ΔSo > 0). Spectroscopic studies (e.g., FTIR, XPS) indicate that the adsorption is based on the formation of inner-sphere complexes between surface active moieties and the uranyl cation. Regeneration and uranium recovery by acidification and complexation using carbonate or chelating ligands (e.g., EDTA) have been found to be successful. The application of aerogel-based adsorbents to uranium removal from industrial processes and uranium-contaminated waste waters was also successful, assuming that these materials could be very attractive as adsorbents in water treatment and uranium recovery technologies. However, the selectivity of the studied materials towards hexavalent uranium is limited suggesting further development of aerogel materials which could be modified by surface derivatization with chelating agents (e.g., salophen, iminodiacetate) presenting high selectivity for uranyl moieties.
2、 Introduction: the last paragraph is too sudden, and some cohesive statements should be added.
our response: The last paragraph of the Introduction has been edited and now reads as follows:
This review presents the recent progress in the development and use of aerogel materials for uranium uptake and recovery from aquatic environments. Knowledge of the chemical behavior and speciation of hexavalent uranium at certain experimental conditions is of fundamental importance to better understand and describe the adsorption processes. In addition, an insight into the physicochemical parameters affecting adsorption efficiency and mechanism will enable the design and development of efficient and selective adsorbents.
3、 Figure 1 is too simple, and the author should enrich it. Figure 5 and Figure 6 should be merged.
our response: Figure 1 is a general scheme showing the main steps along the formation of aerogels. Further expansion will render Figure 1 unnecessarily complicated. Figures 5 and 6 have been merged.
4、 Conclusions: it is a repeated expression with the body content, and it should be reorganized.
our response: Conclusions have been revised and now read as follows:
In the last decade, the number of studies on uranium sorption using aerogels has increased dramatically. This is because of the steadily increasing interest in the production of efficient adsorbent materials for radionuclide removal and recovery from contaminated waters. Thus, captured uranium is planned to cover future demands of the nuclear power industry.
Generally, studies are focused on the effect of different physicochemical parameters (e.g., pH, initial U(VI) concentration, ionic strength, temperature, contact time) on the adsorption efficiency. According to those studies, pH plays a key role because it governs both U(VI) speciation and surface species dissociation and charge.
The isothermal data obtained from experiments related to the effect of metal ion concentration are best fitted by the Langmuir adsorption isotherm model resulting in supreme sorption capacity values (in some cases even above 2000 g of uranium per kg of aerogel) and are the highest ever reported for uranium. The experimental kinetic data indicate fast sorption kinetics (equilibrium is reached from a few minutes to a few hours). In most cases, they are best fitted by the pseudo-second-order kinetic model. Both the Langmuir isotherm and the pseudo-second-order kinetic models are indicative of chemisorption. Other parameters/phenomena, however, such as the diffusion within the porous matrix of aerogels, the adsorbent dose, or the effect of the ionic strength, cannot be ignored. In addition, the associated thermodynamic data generally reveal an endothermic, entropy-driven sorption mechanism, indicating the formation of inner-sphere surface complexes. The formation of inner-sphere complexes between U(VI) and the active groups on the aerogel surfaces is supported mainly by FTIR and XPS data.
Several aerogel materials show excellent performance regarding their reuse even after several adsorption-desorption cycles, with relatively high uranium recovery. However, the presence of multivalent metal cations (e.g., Ca2+, Al3+) and complexing species (e.g., CO32−, PO43−) strongly affect the sorption efficiency of the adsorbents towards uranium, because of competitive sorption and formation of stable solution complexes, respectively, indicating limited selectivity of the studied aerogels towards uranium (U(VI).
Based on this review future studies should focus on the preparation of aerogel materials with specific surface groups possessing high affinity and selectivity towards uranium and other precious and industrial metals/metalloids. This is of particular interest, because, besides sorption affinity and capacity, selectivity is a key factor affecting the recovery of the desired metal from multicomponent and complex industrial processes and wastewater. In addition, in radiopharmaceutical applications, it is of cardinal importance to selectively separate and recover the radionuclide of interest from reaction solutions containing other radionuclides and undesirable by-products.
The most efficient and selective aerogel materials should be tested not only in laboratory settings but also at pilot and industrial scale, in order to attract broader interest and find applications in large-scale/industrial processes. Moreover, data obtained from EXAFS (Extended X-ray absorption fine structure spectroscopy) and TRLFS (Time-resolved laser fluorescence spectroscopy) studies would allow a better understanding of the interaction between U(VI) species and the aerogel surface and describe the mechanism at the molecular level. This is of fundamental importance for the design and development of more effective and selective adsorbents.
5、 The language expression should be polished.
our response: The text has been carefully read and edited, wherever appropriate
Reviewer 2 Report
Aerogels are widely applied in many fields including waste water treatment. The topic of this review is of interesting to broad readers. After carefully reading, it was found that this article needs major revisions because several issues and explanations are still need to be clarified.
1. Waste water treatment is important for the sustainable development. Many absorbents have been developed for heavy metal ions removal. Some typical references are suggested to be cited, e.g. Journal of Bioresources and Bioproducts 2021, 6 (4), 292-322; Journal of Bioresources and Bioproducts 2021, 6 (3), 223-242.
2. A figure summary the whole content of the manuscript is suggested to be added for quick catching the content.
3. What are the advantages of aerogels compared to other nanomaterials in heavy metal removal? Please highlight them in the introduction.
4. A few more figures are suggested to be added to show the synthesis and performance of aerogel based absorbents.
5. There are some interesting conclusions drawn but insufficient focus on future work.
6. Please pay attention to the writing of references. Most of the journal names are written in full name while some are written in abbreviations.
Author Response
Reviewer 2
Aerogels are widely applied in many fields including wastewater treatment. The topic of this review is interesting to broad readers. After carefully reading, it was found that this article needs major revisions because several issues and explanations are still need to be clarified.
our response: the manuscript has been revised and clarifications have been added
- Wastewater treatment is important for sustainable development. Many absorbents have been developed for heavy metal ions removal. Some typical references are suggested to be cited, e.g. Journal of Bioresources and Bioproducts 2021, 6 (4), 292-322; Journal of Bioresources and Bioproducts 2021, 6 (3), 223-242.
our response: The paper “Journal of Bioresources and Bioproducts 2021, 6 (4), 292-322” entitled “Synthesis and Application of Granular Activated Carbon from Biomass Waste Materials for Water Treatment: A Review” is a review paper on biomass-derived materials suitable for water remediation and it has been added in the introduction (new reference 19). The paper “Journal of Bioresources and Bioproducts 2021, 6 (3), 223-242” entitled “New Ulva lactuca Algae Based Chitosan Bio-composites for Bioremediation of Cd(II) Ions” does not report uranium adsorption or the use of any aerogel as a sorbent and it is not within the scope of the present review article. Therefore, this paper has not been added in the list of references.
- A figure summary the whole content of the manuscript is suggested to be added for quick catching the content.
our response: the abstract has been revised and now reads as follows:
Aerogels are a class of lightweight, nanoporous and nanostructured materials with diverse chemical compositions and huge potential for applications in a broad spectrum of fields that have led IUPAC to include them in the top ten emerging technologies in Chemistry for 2022. This review provides an overview of aerogel-based adsorbents that have been used for the removal and recovery of uranium from aqueous environments, as well as an insight into the physicochemical parameters affecting adsorption efficiency and mechanism. Uranium removal is of particular interest regarding uranium analysis and recovery to cover present and future uranium needs for nuclear power energy production. Among the methods used, such as ion exchange, precipitation and solvent extraction, adsorption-based technologies are very attractive due to their easy and low-cost implementation, as well as the wide spectrum of adsorbents available. Aerogel-based adsorbents present extraordinary sorption capacity for hexavalent uranium that can be as high as 8.8 mol kg–1 (2088 g kg–1). The adsorption data generally follow the Langmuir isotherm model, and the kinetic data are in most cases better described by the pseudo-second-order kinetic model. Evaluation of the thermodynamic data reveals that the adsorption is generally an endothermic, entropy-driven process (ΔHo, ΔSo > 0). Spectroscopic studies (e.g., FTIR, XPS) indicate that the adsorption is based on the formation of inner-sphere complexes between surface active moieties and the uranyl cation. Regeneration and uranium recovery by acidification and complexation using carbonate or chelating ligands (e.g., EDTA) have been found to be successful. The application of aerogel-based adsorbents to uranium removal from industrial processes and uranium-contaminated waste waters was also successful, assuming that these materials could be very attractive as adsorbents in water treatment and uranium recovery technologies. However, the selectivity of the studied materials towards hexavalent uranium is limited suggesting further development of aerogel materials which could be modified by surface derivatization with chelating agents (e.g., salophen, iminodiacetate) presenting high selectivity for uranyl moieties.
- What are the advantages of aerogels compared to other nanomaterials in heavy metal removal? Please highlight them in the introduction.
our response: The main advantage related to the specific application is that aerogels generally possess the highest by far adsorption capacities reported.
- A few more figures are suggested to be added to show the synthesis and performance of aerogel-based absorbents.
our response: The aerogels that have been used for uranium adsorption are very different with respect to their chemical composition. Therefore, a figure showing the specific details of the synthetic procedure in each case would be very complicated. Figure 1 is a general scheme showing the main steps along the formation of aerogels. For the synthesis of the aerogel materials, the reader is encouraged to go to the specific original references.
- There are some interesting conclusions drawn but insufficient focus on future work.
our response: Conclusions have been revised and now read as follows:
In the last decade, the number of studies on uranium sorption using aerogels has increased dramatically. This is because of the steadily increasing interest in the production of efficient adsorbent materials for radionuclide removal and recovery from contaminated waters. Thus, captured uranium is planned to cover future demands of the nuclear power industry.
Generally, studies are focused on the effect of different physicochemical parameters (e.g., pH, initial U(VI) concentration, ionic strength, temperature, contact time) on the adsorption efficiency. According to those studies, pH plays a key role because it governs both U(VI) speciation and surface species dissociation and charge.
The isothermal data obtained from experiments related to the effect of metal ion concentration are best fitted by the Langmuir adsorption isotherm model resulting in supreme sorption capacity values (in some cases even above 2000 g of uranium per kg of aerogel) and are the highest ever reported for uranium. The experimental kinetic data indicate fast sorption kinetics (equilibrium is reached from a few minutes to a few hours) and in most cases, they are best fitted by the pseudo-second-order kinetic model. Both the Langmuir isotherm and the pseudo-second-order kinetic models are indicative of chemisorption. Other parameters/phenomena, however, such as the diffusion within the porous matrix of aerogels, the adsorbent dose, or the effect of the ionic strength, cannot be ignored. In addition, the associated thermodynamic data generally reveal an endothermic, entropy-driven sorption mechanism, indicating the formation of inner-sphere surface complexes. The formation of inner-sphere complexes between U(VI) and the active groups on the aerogel surfaces is supported mainly by FTIR and XPS data.
Several aerogel materials show excellent performance regarding their reuse even after several adsorption-desorption cycles, with relatively high uranium recovery. However, the presence of multivalent metal cations (e.g., Ca2+, Al3+) and complexing species (e.g., CO32−, PO43−) strongly affect the sorption efficiency of the adsorbents towards uranium, because of competitive sorption and formation of stable solution complexes, respectively, indicating limited selectivity of the studied aerogels towards uranium (U(VI).
Based on this review future studies should focus on the preparation of aerogel materials with specific surface groups possessing high affinity and selectivity towards uranium and other precious and industrial metals/metalloids. This is of particular interest, because, besides sorption affinity and capacity, selectivity is a key factor affecting the recovery of the desired metal from multicomponent and complex industrial processes and wastewaters. In addition, in radiopharmaceutical applications it is of cardinal importance to selectively separate and recover the radionuclide of interest from reaction solutions containing other radionuclides and undesirable by-products.
The most efficient and selective aerogel materials should be tested not only in laboratory settings, but also at pilot and industrial scale, in order to attract broader interest and find applications in large-scale/industrial processes. Moreover, data obtained from EXAFS (Extended X-ray absorption fine structure spectroscopy) and TRLFS (Time-resolved laser fluorescence spectroscopy) studies would allow a better understanding of the interaction between U(VI) species and the aerogel surface and describe the mechanism at the molecular level. This is of fundamental importance for the design and development of more effective and selective adsorbents.
- Please pay attention to the writing of references. Most of the journal names are written in full name while some are written in abbreviations.
our response: The problem has been fixed.
Round 2
Reviewer 1 Report
The manuscript has been improved greatly and the authors give positive responses to the comments. It can be accepted.
Reviewer 2 Report
The manuscript has been revised according to the comments and could be accepted.